# Can Exposure to Environmental Pollutants Be Associated with Less Effective Chemotherapy in Cancer Patients?

**DOI:** 10.3390/ijerph19042064

**Published:** 2022-02-12

**Authors:** Francisco Alejandro Lagunas-Rangel, Wen Liu, Helgi B. Schiöth

**Affiliations:** 1Department of Surgical Sciences, Functional Pharmacology and Neuroscience, Uppsala University, BMC Box 593, Husargatan 3, 75124 Uppsala, Sweden; wen.liu@neuro.uu.se; 2Institute of Translational Medicine and Biotechnology, I. M. Sechenov First Moscow State Medical University, 8-2 Trubetskaya Str. Moscow, 119991 Moscow, Russia

**Keywords:** chemotherapy resistance, apoptosis, PI3K/AKT pathway, drug efflux, antioxidant enzymes

## Abstract

Since environmental pollutants are ubiquitous and many of them are resistant to degradation, we are exposed to many of them on a daily basis. Notably, these pollutants can have harmful effects on our health and be linked to the development of disease. Epidemiological evidence together with a better understanding of the mechanisms that link toxic substances with the development of diseases, suggest that exposure to some environmental pollutants can lead to an increased risk of developing cancer. Furthermore, several studies have raised the role of low-dose exposure to environmental pollutants in cancer progression. However, little is known about how these compounds influence the treatments given to cancer patients. In this work, we present a series of evidences suggesting that environmental pollutants such as bisphenol A (BPA), benzo[a]pyrene (BaP), persistent organic pollutants (POPs), aluminum chloride (AlCl_3_), and airborne particulate matter may reduce the efficacy of some common chemotherapeutic drugs used in different types of cancer. We discuss the potential underlying molecular mechanisms that lead to the generation of this chemoresistance, such as apoptosis evasion, DNA damage repair, activation of pro-cancer signaling pathways, drug efflux and action of antioxidant enzymes, among others.

## 1. Introduction

Environmental pollutants are compounds that have been released into the ecosystem and can cause health problems for living things exposed to them. It has been suggested that such pollutants are responsible for approximately 9 million deaths per year, which corresponds to 16% of all deaths worldwide. Indeed, three times more deaths than AIDS, tuberculosis, and malaria combined [1].

Since environmental pollutants are ubiquitous and a considerable amount are quite resistant to degradation, we are exposed to many of them on a daily basis. For example, atmospheric air, plastic food and beverage containers, cosmetics, sunscreens, perfumes, cleaning products, and garden products contain pollutants. Thus, these compounds can enter our body through different routes, either in food, by inhalation, or through the skin, among others [2]. Once inside the body, pollutants can bioaccumulate or promote oxidative stress and inflammation, genomic alterations and mutations, epigenetic alterations, mitochondrial dysfunction, endocrine alterations, altered intercellular communication, modify microbiome communities, and impair nervous system function, among many other things [3].

Several epidemiological findings, along with a better understanding of the mechanisms that link toxic substances to disease development, suggest that exposure to some environmental pollutants, even at low or very low concentrations, may lead to an increased risk of developing cancer and/or to accelerate its progression [4,5]. In this sense, it is interesting that the Asian continent, which within its geographic extension contains the cities with the highest pollution rates [6], is also the continent with the highest incidence (49.3% of cases worldwide) and mortality rates (58.3% of cases worldwide) due to cancer [7]. This relationship could be explained since certain environmental pollutants can promote sustained growth by affecting the cell cycle; prevent apoptosis by avoiding the activation of intrinsic and/or extrinsic pathways; promote angiogenesis by facilitating the secretion of factors such as VEGF, FGF, and TNF-α; activate invasion and metastasis by stimulating the epithelial-to-mesenchymal transition (EMT) and the secretion of metalloproteases; and promote genomic instability and chronic inflammation, among other ways [8,9,10].

Although a great deal of knowledge has been generated in recent years about the role of environmental pollutants in cancer, little is known about the effects these compounds have on the treatment of cancer patients and, more importantly, if they are related to a lower efficacy of chemotherapeutic drugs and the generation of chemoresistance. Chemoresistance is the ability of cancer cells to evade or cope with the presence of therapeutic agents and, currently, it represents one of the main challenges for the treatment of cancer [11]. Our hypothesis is that environmental contaminants could affect the effectiveness of chemotherapeutic treatments, even at concentrations below those established by current regulations, and thus negatively affect the clinical prognosis of cancer patients. In this way, the objective of this work is to show evidence that suggests that environmental pollutants can reduce the efficacy of chemotherapeutic drugs (Table 1), with a special emphasis on explaining the possible underlying molecular mechanisms that lead to the generation of this chemoresistance.

## 2. Methods

This review is based on evidence collected by performing a PubMed and Google Scholar query using the words “cancer AND resistance AND pollutants” as search terms. The search strategy was implemented by manually searching the references reported by the most relevant studies on this topic. Figure 1 shows a flowchart of the steps that were followed to find evidence that contaminants could affect the effectiveness of chemotherapy drugs, as well the number of articles excluded and included in each step based on the Preferred Reporting Items for Systematic Reviews and Meta-Analyses (PRISMA 2020).

## 3. Mechanisms of Action of Chemotherapy Drugs

Many different types of chemotherapy drugs are used to treat cancer, either alone or in combination with other drugs or treatments. These drugs are very different in their chemical makeup and mechanism of action (Figure 2), how they are prescribed and given, how helpful they are for treating certain types of cancer, and the side effects they can have [23,24]. In order to explain how environmental pollutants can alter the effects of chemotherapeutic drugs, it is necessary to understand the mechanisms of action of the latter. Thus, below we present a brief description of the mechanisms used by those drugs of relevance in the following parts of this work and commonly used in different types of cancer such as cisplatin (CDDP), doxorubicin (DOX), 5-fluorouracil (5-FU), vinblastine (VIN), vincristine (VCR), paclitaxel (PTX), camptothecin (CPT), and tamoxifen (TAM).

CDDP is one of the most potent antitumor agents known and shows clinical activity against a wide variety of solid tumors. Its cytotoxic mode of action is mediated by the generation of DNA adducts, particularly intrastrand crosslink adducts, which activate several signal transduction pathways, including those involving ATR, p53, p73, and MAPK that lead to the activation of apoptosis [25]. DOX is classified as an anthracycline antibiotic and is commonly used to treat some hematologic malignancies, such as leukemias and Hodgkin’s lymphoma, as well as solid tumors, such as cancers of the bladder, breast, stomach, lung, ovaries, thyroid, soft tissue sarcoma, and others. Its main mechanism of action is through its intercalation in DNA and the disruption of topoisomerase-II-mediated DNA repair [26]. In addition, DOX can also induce ROS production, damage mitochondrial DNA, disrupt major mitochondrial functions, and reduce membrane potential with the consequent release of cytochrome C and induction of apoptosis, among other things [27]. Meanwhile, 5-FU is an antimetabolite analogous to uracil but with the difference that it has a fluorine atom in the C-5 position instead of hydrogen. This drug is widely used in the treatment of a variety of cancers, including colorectal, breast, and aerodigestive tract cancers, where it exerts its anticancer effects by inhibiting thymidylate synthase (TS) and incorporating its metabolites into the RNA and DNA [28]. VIN and VCR are vinca alkaloids that specifically bind to tubulin and block its polymerization to form microtubules. As a consequence of this, a spindle poisoning occurs that causes the chromosomes to disperse and subsequently leads to cell death. These are used in hematological neoplasms, such as Hodgkin and non-Hodgkin lymphoma, as well as in solid tumors, such as lung cancer, neuroblastoma, rhabdomyosarcoma, osteogenic sarcoma, bladder cancer, among others [29]. In contrast to VIN and VCR, PTX is a microtubule stabilizing drug that induces mitotic arrest and/or multipolar divisions that trigger cell death. It is approved for the treatment of ovarian, breast, and lung cancer, as well as Kaposi’s sarcoma [30]. CPT is a plant alkaloid derived from the *Camptotheca acuminata* tree. This compound binds to topoisomerase I and stabilizes its interaction with DNA, preventing the separation of this complex and thus inhibiting enzyme activity. An irreversible arrest of the replication fork and breaks in the DNA are caused that eventually trigger the arrest of the cell cycle in the G2 phase and cell death [31]. Currently, its derivatives have become standard components in the treatment of various neoplasms, such as ovarian cancer, small cell lung cancer, colorectal cancer, and malignant neoplasms of the upper gastrointestinal tract, among others [32]. Finally, TAM is a selective estrogen receptor (ER) modulator and is the most widely used drug for the treatment of estrogen receptor positive breast cancer. It prevents the activation of the ER signaling pathway by competing with its ligand 17β-estradiol (E2) [33]. In addition, many chemotherapeutic agents can also induce the overproduction of reactive oxygen species (ROS) to kill cancer cells (by exceeding the cytotoxic threshold). ROS damage cell membranes, DNA, and proteins and consequently activate regulated cell death programs that mainly include apoptosis, necroptosis, and ferroptosis [34].

## 4. Molecular Mechanisms of Chemoresistance in Cancer

Currently, several molecular chemoresistance mechanisms have been identified, which mainly include an enhanced efflux of drugs, oncogenic signaling pathways, apoptosis avoidance, tumor microenvironment signals, down-regulation of tumor suppressor genes, mitochondrial alteration, activation of antioxidant enzymes and scavengers, increased DNA repair capacity, increased metabolism of xenobiotics, autophagy, EMT, and an increase in cancer stem cells (CSCs). Notably, more than one mechanism can be used by the cancer cell to generate chemoresistance [35,36]. Below, we give a brief description of those that we consider relevant to better understand the following parts of this review.

### 4.1. Enhanced Drug Efflux

One of the main causes of drug resistance is increased drug efflux by ATP-binding cassette (ABC)-superfamily multidrug efflux pumps, which often consist of multiple subunits and whose function is to translocate substrates through the membrane against the concentration gradient using the energy released by the hydrolysis of ATP in its subunits with ATPase activity. A total of 15 members of this protein family have been implicated in potentially conferring resistance to chemotherapeutic agents, however three of them are the most studied, being the P-glycoprotein (P-gp/MDR1/ABCB1), multidrug resistance protein1 (MRP1/ABCC1), and breast cancer resistant protein (BCRP/ABCP/MXR) [37,38]. P-gp overexpression has been observed in different types of hematological and solid tumors, where it favors the discharge of drugs, such as colchicine, tacrolimus, dexamethasone, steroids, quinidine (QUIN), etoposide (VP-16), DOX, and VIN [35]. Meanwhile, MRP1 prevents the intracellular accumulation of glutathione (GSH)-conjugated xenobiotics and GSH-conjugated metabolites, which also confers drug resistance in tumor cells treated with daunorubicin, methotrexate (MTX), DOX, VP-16, and VCR [39]. Ultimately, BCRP is overexpressed in breast cancer and has a negative correlation with intracellular levels of mitoxantrone, daunorubicin, topotecan, and DOX [35].

### 4.2. Activation of Oncogenic Signaling Pathways

The PI3K/AKT pathway is an important signaling pathway for chemoresistance in a variety of cancers including breast cancer, leukemia, lung cancer, ovarian cancer, hepatocellular carcinoma, and melanoma. This pathway generates a survival signal to resist cytotoxic anticancer drugs and improves the characteristics of cancer stem cells (CSCs) [40]. The MAPK/ERK pathway is widely used by cancer cells subjected to environmental stresses such as chemotherapy and the antitumor activity of the host immune system. Chemoresistance occurs because this pathway cooperates with some factors of the tumor microenvironment, activates pro-survival pathways that lead to cell proliferation and migration, as well as with the positive regulation of drug efflux transporters and the modulation of apoptosis, differentiation, and senescence [41]. Meanwhile, activation of the NF-κB pathway has been reported with various chemotherapeutic agents such as daunomycin, bortezomib, PTX, VIN, VCR, DOX, 5-FU, CDDP, and TAM. This pathway frequently contributes to the induction of chemoresistance and radioresistance through the regulation of several genes involved in immunoregulation, growth regulation, inflammation, carcinogenesis, and apoptosis [42]. Notch signaling is frequently deregulated in many cancers, most commonly by over-activation, and confers a survival advantage on tumors. Recent studies show that Notch signaling favors chemoresistance by promoting the characteristics of CSCs and inducing EMT [43]. The NRF2-ARE pathway regulates GSH metabolism and the expression of enzymatic antioxidant systems and their cofactors to restore redox homeostasis [44]. Several studies have shown that cancer cells with high levels of NRF2 are less sensitive to etoposide, CDDP, and DOX, since there are more antioxidants that counteract the ROS produced by these drugs [45]. Furthermore, NRF2 has also been associated with sustained proliferative signaling and insensitivity to anti-growth signals, resistance to apoptosis, sustained angiogenesis, tissue invasion and metastasis, metabolic reprogramming, and immune system evasion [44]. RAS is the most frequently mutated oncogene in human cancers, and KRAS is the most frequently mutated RAS isoform. The mutated RAS protein is constitutively activated and enhances signaling pathways that promote cell growth, proliferation, and survival, as well as mobility and migration [46]. Oncogenic KRAS has been reported to be capable of transcriptionally increasing NRF2 levels and enhancing chemoresistance [47]. 

### 4.3. Increased DNA Repair Capacity

Another possibility that tumor cells become resistant to a variety of anticancer drugs is their ability to repair DNA damage. Cancer cells can overcome DNA damage caused by chemotherapeutic drugs by activating several different repair mechanisms and pathways [48,49]. For example, if the repair pathway that is responsible for triggering cell death after chemotherapy therapy is deficient, an alternative repair pathway compensates and prompts cancer cells to repair the damage, leading to resistance to chemotherapy [50]. A significant correlation has been shown between overexpression of the XPF and ERCC-1 proteins of the nucleotide excision repair (NER) pathway and the development of CDDP resistance in cancer cells [36]. Furthermore, the repair and tolerance of these lesions also involves homologous recombination (HR) pathway proteins such as RAD51, MRE11, ATM, ATR, and BRCA1 [36,51]. Defects in the ATM-Chk2-p53 pathway also contributed to the radiation resistance of glioblastoma cells [52]. Meanwhile, DNA-PK activity, involved in double-strand repair through the non-homologous end joining (NHEJ) pathway, plays a role in chemoresistance and its inhibition enhances the sensitivity of cancer cells to different chemotherapeutic drugs [53]. O6-methylguanine-DNA methyltransferase (MGMT) overexpression has contributed to an acquired resistance to temozolomide (TMZ) and other alkylating agents [54]. Meantime, the process of mutagenic translesion synthesis (TLS), responsible for the repair of inter-strand DNA cross-links, has also been significantly associated with the development of resistance in cancer cells [36].

### 4.4. Elevated Xenobiotic Metabolism

Most chemotherapeutic drugs are subject to being metabolized by cytochrome P450. Cytochrome isoenzymes such as CYP1A6, CYP1A2, CYP1B1, CYP2C9, CYP2B6, CYP2C19, CYP3A4/5, and CYP2D6 are essential for phase I drug metabolism and detoxification [55]. CYP1B1 overexpression in various types of cancer has previously been reported to modify the biotransformation of chemotherapeutics, such as mitoxantrone, flutamide (FLUT), docetaxel (TXT), and PTX [56]. In addition, increased expression of the CYP2A6 enzyme, which is involved in the metabolism of ifosfamide (IFO), cyclophos-phamide (CTX), aflatoxin, and 5-FU, has been found in some tissues of chimeric resistant breast tumors [57]. Furthermore, the highly upregulated expression of CP4Z1, CYP1B1, and CYP2A7 in cancer cells was associated with their increased resistance to a variety of chemotherapeutic agents [58].

### 4.5. Increased CSCs

CSCs are a cellular subpopulation of cancer cells characterized by their ability to self-renew, multiple differentiation, drug resistance, and tumor formation and growth [35]. The chemoresistance of CSCs may be related to the fact that these cells maintain a quiescent state, activate drug afflux mechanisms, have enhanced DNA repair mechanisms, and can acquire an EMT phenotype, among many other things [59]. Thus far, it is unknown whether the number of CSCs in a tumor increases as a result of selection due to chemotherapeutic agents or because conditions are propitiated for these cells to proliferate, or both [60]. However, when the number of CSCs increases, it is associated with a worse clinical prognosis, a more aggressive phenotype and chemoresistance [61].

### 4.6. Extracellular Matrix (ECM)

The tumor microenvironment comprises several components that are not malignant by themselves, however play a fundamental role in creating adequate conditions for the growth and sustainability of tumors, such as tumor vasculature, connective tissue, infiltrating immune cells, and the extracellular matrix (ECM), among others [62]. ECM and its sequestered growth factors are a fundamental component for all cells, although its role in CSCs has recently been highlighted. Pathologic remodeling of the ECM is an established hallmark of cancer, and the ECM is a key mediator of metastasis and drug resistance [63]. Several mechanisms of chemoresistance involving the ECM have been identified in all types of cancer and have been classified into a variety of categories including physical barriers to treatment (hypoxia, pH, and interstitial fluid pressure), associated drug resistance with cell adhesion (ECM organization, mechanosignaling, and pro-survival signaling pathways) and the effect of ECM on subpopulations of inherent stem cells or CSCs specifically [62,63]. Furthermore, ECM may also have implications for other cellular mechanisms that promote resistance to chemotherapy, such as DNA repair and oxidative stress, among many others [64,65].

## 5. Evidence of Pollutants Affecting the Efficacy of Chemotherapeutic Drugs

### 5.1. Bisphenol A (BPA)

BPA is a plasticizer widely used in products intended for direct contact with food and sanitary consumables, including plastic containers, kitchen utensils, inner liners of cans and jar lids, as well as medical equipment, steel drums, and pipes. For this reason, we are very prone to ingesting foods and beverages contaminated with this compound [66]. Moreover, BPA is an endocrine disruptor that has been shown to have important health effects, including the development and progression of different types of cancer [67]. In the normal lung fibroblast cell line MRC-5, it was observed that BPA can mitigate the cytotoxic effects of DOX. In these experiments, cells were pre-treated for 24 h with BPA (in concentrations between 0.44 nM–4.4 µM) and then a combined treatment of DOX (4 µM) and the pollutant was administered for another 24 h. Thus, when the viability of cells exposed only to the chemotherapeutic agent and those treated with BPA and DOX was compared, a reduction in cell death was found in the latter, which was associated with less oxidative stress and DNA damage, as well as a greater number of micronuclei [15]. Probably also involved in these findings and has been previously described, BPA can promote a higher expression of several DNA repair proteins such as ATM, BRCA1, RAD51, RAD 50, CtIP, MRE11A, XRCC6, BARD1, SMC1A, PRKDC, and BRCC3 [68,69] and also cause chromatin compaction with a consequent reduction in the amount of topoisomerase I-DNA covalent complexes and DNA strand breaks [16]. Thus, BPA could reduce DNA damage caused by DOX or another chemotherapeutic agent. In another study, pre-treatment with BPA at nanomolar doses (0.1–10 nM) for 24 h in breast cancer cells, both estrogen-sensitive and non-sensitive, followed by a conjoint treatment for another 24 h of DOX, CDDP, or vinblastine (VIN) with the pollutant gave cells greater chemoresistance [12]. Indeed, this chemoresistance was comparable to that caused by estradiol [13]. Particularly in the case of the CDDP-BPA and DOX-BPA co-treatments, elevated levels of the anti-apoptotic proteins BCL-2 and BCL-xL were found with respect to treatments only with chemotherapeutic drugs [12,13]. In HT29 colon adeno-carcinoma cells, joint treatment of BPA (at a concentration in accordance with the established limit) with DOX (4.4 µM) did not produce significant effects, however when the cells were pre-treated for 24 h with BPA and subsequently combined treatment with both compounds was administered for another 24 h, it resulted in a reduction of apoptotic bodies and changes in gene expression, thus avoiding the overexpression of the cell cycle regulatory genes AURKA, CDKN1A, and CLU and the reduction of the expression of the c-FOS gene [14]. Finally, a study in MCF7 breast cancer cells was presented at the EUROTOX 2019 Congress that reported a decrease in the cytotoxic effects of VCR (5.45 nM) and TAM (9 nM) after preincubation for 4 h with BPA or bis (2-ethylhexyl) phthalate (DEHP) at nanomolar concentrations (0.1–100 nM) [70]. In this case, BPA can act as an estrogen agonist that could compete with the antiestrogenic effect of TAM [71] and has also been shown to promote microtubule polymerization and centrosome-based microtubule nucleation, which are the opposite effects of VCR [72].

### 5.2. Benzo[a]pyrene (BaP)

BaP is a ubiquitous polycyclic aromatic hydrocarbon (PAH) that results from incomplete combustion of organic matter and can be found in coal tar, automobile exhaust, tobacco smoke, and charbroiled food [73]. BaP has an impact on the initiation, promotion, and progression of cancer, acts as a genotoxic and non-genotoxic carcinogen by forming DNA adducts, and also activates AhR receptors, among many other things [74]. BEAS-2B bronchial epithelial cells transformed with BaP showed greater resistance to DOX, high levels of stem cell-like markers, and overexpression of proteins involved in resistance to anticancer drugs such as ALDH1A1, ABCG2, SOX2, c-MYC, and KLF4 [75]. Meanwhile, co-treatments of BaP (10 μM) with CDDP (4.2 μM), 5-FU (3.5 μM), or PTX (2 μM) for 24 h in oesophageal cancer WHCO1 and WHCO5 cells caused a decrease in drug-induced apoptosis and an increase in the number of colonies when compared to cells treated with chemotherapy drugs alone. These same effects were also found when BaP was administered together with CDDP-5-FU, CDDP-PTX, or 5-FU-PTX combination therapies. Interestingly, when BaP was administered with a single chemotherapeutic drug, the expression of CYP1A1 and CYP1A2 increased, while when it was administered with the combination of drugs, the expression of these genes decreased, as well as that of CYP1B1 and GSTP1. BaP was also reported to increase the activation of the MEK/ERK and PI3K/AKT pathways, migration, and cell invasion when administered in conjunction with the drug combination [17]. Furthermore, exposure to BaP has been reported to induce overexpression of the ATP-dependent P-gp efflux pump, which could cause drugs to be expelled from the cell prior to performing their function [76]. In another study, squamous cell carcinoma (SCC) CAL27 and SCC9 cells treated with long-term (3 months) BaP (50 nM) showed greater chemoresistance to 5-FU (≤100 µg/mL) and CDDP (≤100 µM). Although no mechanism was analyzed to explain chemoresistance, it was mentioned that these cells also exhibited greater mobility, invasiveness, and aggressiveness, presenting shortened telomeres and diverse genomic mutations [18].

### 5.3. Persistent Organic Pollutants (POPs)

POPs are a class of carbon-based organic chemicals that are difficult to break down and therefore persist and bioaccumulate in the environment. Examples of these are pesticides, especially organochlorine pesticides (OCP) such as dichlorodiphenyltrichloroethane (DDT) and their metabolites; industrial and technical chemicals, including polychlorinated biphenyls (PCBs), polybrominated diphenyl ethers (PBDEs), and perfluorooctane sulfonate (PFOS); and by-products of industrial processes including polychlorinated dibenzo-p-dioxins (PCDD) and polychlorinated dibenzofurans (PCDF), among others [77]. Exposure to these pollutants can cause various health problems such as endocrine disorders, cardiovascular diseases, obesity, diabetes, reproductive and neurological ailments, learning disabilities, and cancer [78]. In this way, pre-treatment for 24 h with low doses of PCB-1254 (1 μg/mL) or hexabromocyclododecane (HBCD, 1 μM) followed by a combined treatment of either of these compounds and CDDP (10 μM) for another 24 h reduced chemotherapeutic-mediated cell death by at least 5% in hepatocellular carcinoma cells regardless of whether they were associated with the hepatitis virus or not. In this regard, it was found that both pollutants activate the PI3K/AKT and NF-κB pathways, and increase MDM4 levels with the consequent decrease in p53 [21]. Although not analyzed in this study, pretreatment with PCB-1254 has been reported to improve the DNA repair response in rat hepatocytes [79,80], while HBCD increases the levels of the DNA repair proteins BRCA1, ATM, OGG1, and MTH1 [81]. Thus, improvements in DNA repair systems by these two compounds could collaborate to attenuate the cytotoxic effects of CDDP. Another study showed that prolonged treatment (2 months) with low doses of HBCD (0.0015 nM) significantly reduced the effects of DOX in HME1 cells, which are normal mammary epithelial cells immortalized with the retrovirus pBabepuro+hTERT vector [22]. Prolonged exposure to HBCD has also been shown to increase levels of cyclin E and CDK4, while reducing those of p21 and BAX [82,83]. On the other hand, long-term treatment (2 months) with low doses of 4-tert-octylphenol (OP, 0.0048 nM) reduced the effects of DOX significantly in HME1 cells, although in this study no related mechanism was analyzed [22]. However, possibly the fact that OP can activate the MAPK/ERK pathway and interrupt the innate immune response at very low concentrations may be involved in these findings [84].

### 5.4. Aluminum Chloride (AlCl_3_)

Epidemiological and experimental findings indicate that aluminum is not as harmless as previously thought and that it may contribute to the development of disease [85]. Aluminum (Al^+3^), mainly as chloride salts, can contaminate food, water, beverages, clothing, and cosmetics since it is widely used as a component of food additives, vaccines, antacids, deodorants, parenteral fluids, and kidney dialysates [86]. Indeed, chronic exposure to low concentrations of aluminum chloride (AlCl_3_) has been shown to play a potential role in carcinogenesis [87] and may promote tumorigenesis and metastasis [88]. In HepG2 cells, pre-treatments for 48 h with AlCl_3_ (≤200 μM) followed by combined treatments of the contaminant and 5-FU (100 μM) for another 48 h caused a AlCl_3_ concentration-dependent reduction in apoptosis. This phenomenon was explained by finding a higher concentration of the anti-apoptotic protein BCL-xL and a lower concentration of the pro-apoptotic protein BAX in the cells with the co-treatments than in those treated only with the drug. Furthermore, there was also a reduction in the concentration of reactive oxygen species (ROS) and an increase in the levels of the antioxidant enzymes GPx-1 and SOD2, both also dependent on the concentration of the pollutant. Importantly, it was observed that co-treatments caused the activation of the MAPK/ERK signaling pathway and as a result of this, the progression of the cell cycle was promoted by increasing the phosphorylation of CHK2 in threonine 68, as well as cell migration increased as a result of the high levels of metalloproteases MMP-2 and MMP-9 [19]. Presumably, the antioxidant activity of AlCl_3_ may also have some connection with these findings since it could mitigate ROS induced by chemotherapeutic agents to some degree [89].

### 5.5. Airborne Particulate Matter (PM)

Airborne PM consists of a heterogeneous mixture of solid and liquid particles suspended in the air that continuously vary in size and chemical composition in space and time. Thus, these can contain nitrates, elemental and organic carbon sulfates, organic compounds (e.g., PAHs), biological compounds (e.g., endotoxin, cell fragments), and metals (e.g., iron, copper, nickel, zinc, and vanadium) [90]. Exposure to PM has been identified as the cause of numerous health effects, including respiratory and cardiovascular diseases, reproductive and central nervous system dysfunctions, and cancer [91]. In BEAS-2B cells transformed with Ad12-SV40 2B, it was found that prolonged exposure (5 weeks) to fine particles of 2.5 μm or smaller (PM2.5, biomass combustion product) decreases the cytotoxic effects of DOX (1 μM) by preventing its intracellular accumulation. This phenomenon was related to an increase in GSH levels and a consequent up-regulation of the drug efflux transporter MRP2 (which uses GSH conjugates as a substrate) [20]. Another aspect that could also be collaborating to generate chemoresistance is that chronic exposure to PM2.5 promotes an increase in cells with properties of cancer stem cells (CSCs) through the activation of the Notch signaling pathway and changes in the levels of stemness-associated miRNAs [92,93]. Furthermore, PM2.5 can modify the extracellular matrix and this could also favor chemoresistance. For example, after 5 weeks of exposure to PM2.5, bronchial epithelial cells were significantly enriched in genes associated with extracellular matrix organization, while the levels of genes related to cell adhesion decreased [94].

## 6. Conclusions

We are not aware of other works that have searched and compiled articles related to how exposure to environmental contaminants may be associated with less effective chemotherapy in cancer patients. Our study resulted in several different molecular mechanisms being involved. In general, the literature is scattered and addresses different topics on how contaminants can cause chemoresistance. A molecular mechanism highlighted here is the avoidance of apoptosis by promoting an antiapoptotic context. This is related to the activation of signaling pathways such as PI3K/AKT, ERK/MAPK, and NF-κB that favor survival, cell cycle progression, proliferation, inflammation, migration, and invasion; abolition of p53 signaling; mitigation of DNA damage through chromatin compaction and better repair of these; as well as drug efflux and the action of antioxidant enzymes (Figure 3). It was also mentioned that environmental pollutants have effects on the ECM and these, in turn, could collaborate in chemoresistance, although experiments are necessary to establish this association. All these mechanisms work together in such a way that cells acquire resistance to chemotherapeutic agents and the incidence of chemoresistance increases in different types of cancer, so that treatments require an increase in dose or a change in drug to be effective; most of the time, to more aggressive treatment protocols and medications for patients.

Notably, all the experiments that we analyze here have been carried out only in cell lines, which have not always been able to predict real responses due to model failures, such that these no longer preserve the tumor heterogeneity present in primary cancer and that neither contain the relevant components of the tumor microenvironment, among others. Furthermore, the bioavailability of pollutants may be different in each tissue and organ, even being absent in some of them due to the presence of physiological barriers (for example, the blood-brain barrier, nasal barrier, dermal barrier, intestinal barrier, etc.) or because the levels of these compounds are greatly reduced by the first pass effect. However, this work shows that there is growing evidence that environmental pollutants can affect and reduce the efficacy of chemotherapeutic drugs, which then needs to be confirmed in *in vivo* models and also in clinical trials. Currently, relatively little attention is paid to background contamination during drug treatment and this is not taken into account for dosage setting or regulatory purposes. In general, we know little about the effects of prolonged exposures to contaminants on drug effects at concentrations below the threshold considered safe, and effects on cell lines have been demonstrated. More standardized and integrated experiments are needed before such interactions can affect rules or legislation, which is likely to happen in the future.

## Figures and Tables

**Figure 1 ijerph-19-02064-f001:**
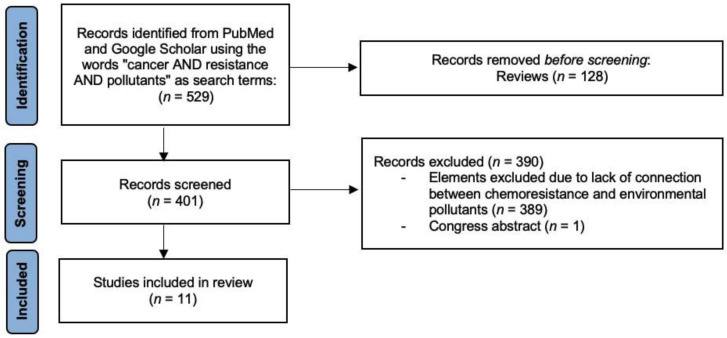
Flowchart describing the protocol adopted in this review based on PRISMA 2020.

**Figure 2 ijerph-19-02064-f002:**
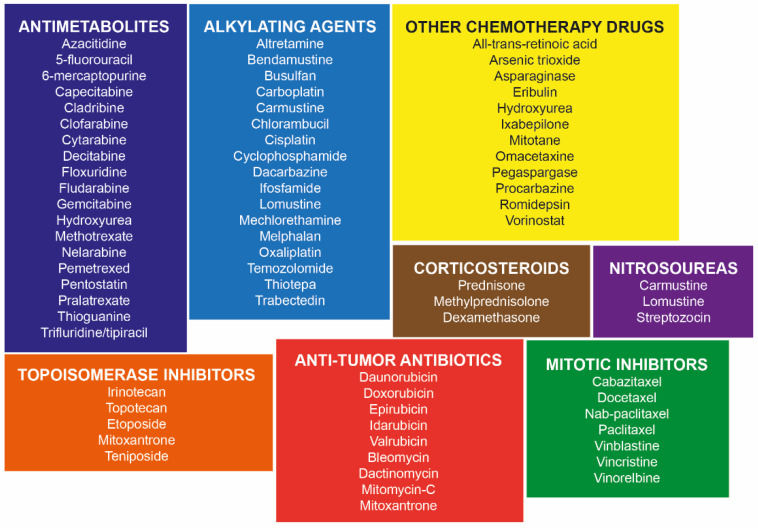
Classification of chemotherapeutic agents based on their mechanism of action.

**Figure 3 ijerph-19-02064-f003:**
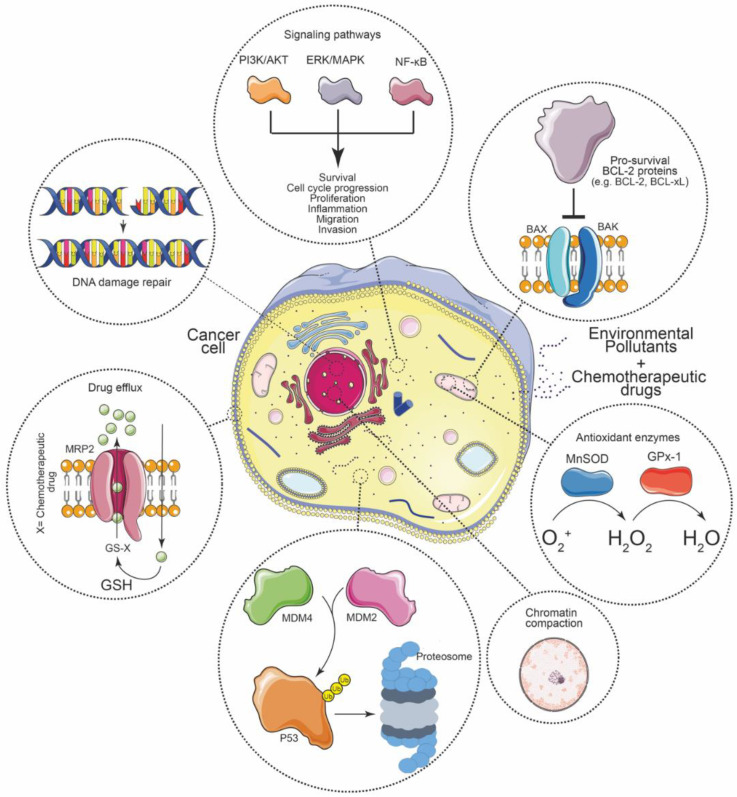
Probable mechanisms involved in reducing the effectiveness of chemotherapeutic agents due to environmental pollutants.

**Table 1 ijerph-19-02064-t001:** Environmental pollutants that have been shown to affect the effectiveness of chemotherapeutic drugs. The results that were only presented at congresses are not shown, although these are detailed in text. DN(M)EL: Derived no-effect or minimum effect level, BPA: Bisphenol A, BaP: Benzo[a]pyrene, AlCl3: Aluminum chloride, PCB-1254: Polychlorinated biphenyls, HBCD: Hexabromocyclododecane, OP: 4-tert-octylphenol, CDDP: Cisplatin, DOX: Doxorubicin, VIN: Vinblastine, 5-FU: 5-fluorouracil, PTX: Paclitaxel, CPT: Camptothecin, HCC: Hepatocellular carcinoma, BC: Breast cancer, COAD: Colon adenocarcinoma, OC: Oesophageal cancer, SCC: Squamous cell carcinoma, MEF: Mouse embryonic fibroblasts, and CNM: Concentration not mentioned.

Pollutant	DN(M)EL Long-Term Exposure			Concentration Tested	Chemotherapeutic Drug	Cancer	Model	Mechanism Associated with Decreased Efficacy of the Chemotherapeutic Drug	Reference
	Inhalation ^a^	Dermal ^b^	Oral ^b^						
BPA	---	0.042 μg ^c^	4 μg ^c^	1 nM/48 h	CDDP ≤400 ng/mL/24 h	BC	T47D cells MDA-MB-468 cells	- Increase in BCL-2 and BCL-xL levels	[12,13]
				1 nM/48 h	DOX ≤125 ng/mL/24 h	BC	T47D cells MDA-MB-468 cells	- Increase in BCL-2 and BCL-xL levels	[12]
				1 nM/48 h	VIN 1 ng/mL/24 h	BC	T47D cells MDA-MB-468 cells	---	[12]
				4 μM/48 h	DOX 4.4 μM/24 h	COAD	HT29 cells	- Prevented an increase in the expression of the AURKA, CDKN1A, and CLU genes - Avoided a reduction in the expression of the c-FOS gene - Reduction in the number of apoptotic bodies	[14]
				≤4.4 µM/48 h	DOX 4 μM/24 h	---	MRC-5 cells	- Less oxidative stress and DNA damage - Greater number of micronuclei	[15]
				150 μM/24 h	CPT 100 nM/24 h	---	MEF cells	- Greater compaction of chromatin - Reduction in the amount of topoisomerase covalent complexes	[16]
BaP	1.43 µg ^c^	---	0.5 µg ^c^	10 μM/24 h	CDDP 4.2 μM/24 h	OC	WHCO1 cells WHCO5 cells	- Increased expression of CYP1A1 and CYP1A2 genes	[17]
				10 μM/24 h	5-FU 3.5 μM/24 h	OC	WHCO1 cells WHCO5 cells	- Increased expression of CYP1A1 and CYP1A2 genes	[17]
				10 μM/24 h	PTX 2 μM/24 h	OC	WHCO1 cells WHCO5 cells	- Increased expression of CYP1A1 and CYP1A2 genes	[17]
				10 μM/24 h	CDDP + 5-FU 4.2 μM + 3.5 μM/24 h	OC	WHCO1 cells	- Activation of the MEK/ERK and PI3K/AKT pathways	[17]
				10 μM/24 h	CDDP + PTX 4.2 μM + 2 μM/24 h	OC	WHCO1 cells	- Activation of the MEK/ERK and PI3K/AKT pathways	[17]
				10 μM/24 h	5-FU + PTX 3.5 μM + 2 μM/24 h	OC	WHCO1 cells	- Activation of the MEK/ERK and PI3K/AKT pathways	[17]
				50 nM/3 months	CDDP ≤100 μM/48 h	SCC	CAL27 cells SCC9 cells	---	[18]
				50 nM/3 months	5-FU ≤100 μg/mL/48 h	SCC	CAL27 cells SCC9 cells	---	[18]
AlCl_3_	4 mg ^c^	2.32 mg ^c^	2.3 mg ^c^	≤200 μM/96 h	5-FU 100 μM/48 h	HCC	HepG2 cells	- Increased BAX and BCL-xL levels - Less ROS production and higher concentration of GPx-1 and SOD2 - Activation of the ERK/MAPK signaling pathway - Greater phosphorylation of CHK2 in Thr68 - Higher levels of MMP-4 and MMP-9	[19]
PM_2.5_	25 μg ^d^	---	---	100 μg/mL/5 weeks	DOX 1 μM/48 h	---	BEAS-2B cells	- Low intracellular accumulation of DOX - Increased levels of GSH - Positive regulation of MRP2 activity	[20]
HBCD	719 μg ^c^	1020 mg ^c^	102 μg ^c^	1 μM/48 h	CDDP 10 μM/24 h	HCC	HepG2 cells MHCC97H cells	- Activation of the PI3K/AKT and NF-κB pathways - Increased levels of MDM4 - Decreased levels of p53	[21]
				0.0015 nM/2 months	DOX CNM/12 h	---	HME1 cells	---	[22]
PCB-1254	---	---	---	1 μg/mL/48 h	CDDP 10 μM/24 h	HCC	HepG2 cells MHCC97H cells	- Activation of the PI3K/AKT and NF-κB pathway. - Increased levels of MDM4 - Decreased levels of p53	[21]
OP	0.6 mg ^c^	5.6 mg ^c^	0.1 mg ^c^	0.0048 nM/2 months	DOX CNM/12 h	---	HME1 cells	---	[22]

^a^ Per cubic meter, ^b^ Kg body weight/day, ^c^ European chemical agency (ECHA) https://echa.europa.eu (accessed on 15 January 2021), ^d^ European environment agency (EEA) https://www.eea.europa.eu (accessed on 15 January 2021).

## Data Availability

All the information described here can be consulted in the respective articles that are shown in the references.

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
