# Peer review of "Can Exposure to Environmental Pollutants Be Associated with Less Effective Chemotherapy in Cancer Patients?"

_ijerph, 2022, doi:10.3390/ijerph19042064_

Round 1

Reviewer 1 Report

The manuscript “Can exposure to environmental pollutants be associated with less effective chemotherapy in cancer?” provides an interesting overview of the literature establishing a possible link between environmental pollutants and chemotherapy resistance. The text is clearly written.

It would be important that the authors clarify why would one look for the link between environmental pollutants and chemotherapy resistance. A clear hypothesis for the analysis presented in this manuscript would help the reader to understand the reasoning behind this extensive literature review. In addition, the authors highlight some specific chemoresistance mechanisms that the authors “consider relevant” (line 129), but this should be better explained. The pathways discussed within each of these specific chemoresistance mechanisms are limited, taking into account the vast literature for each on of these. For example in the topic “Activation of oncogenic signaling pathways”, keys pathways/key factors such as KRAS or EGFR. In particular the first one has been linked to NRF2 activation, mentioned by the authors (PMID: 30824678). Similarly in the DNA repair topic, some key pathways such as NHEJ are missing (see for example PMID: 33628188) or for the xenobiotics metabolism (see for example PMID: 26180516). Also it would be important to discuss the role of extracellular matrix (ECM), both as part of tumor microenvironment and EMT. ECM plays a critical role in chemotherapy resistance (PMID: 32118030), being also affected by environmental pollutants (PMID: 31354480). Moreover it can have implications for other cell mechanisms that promote chemotherapy resistance, such as DNA repair, oxidative stress or cell signalling (PMID: 34394183, PMID: 34733852). Specific ECM alterations occur in response to environmental pollutants and examples should be included in the discussion part.

Specific comments

- The headings Material and Methods, Results and Discussion should be replaced but other more specific headings such as “chemotherapeutic agents”, “chemoresistance mechanisms” and . The headings in their current form are not related with the content of each section

- Table 1 – Legend for the letters “a”, “b” and “c” is missing

- Lines 91-93 – the mode of action of DOX may be through direct TopoII targeting or ROS generation. In addition the role of TopoII os much broader then what has been described

- Lines 122-123 – the sentence “identified, but mainly these include” could be changed into “identified, which mainly include” The authors provide an extensive list of possible mechanisms

Author Response

Dear Reviewer,

We appreciate all your comments that help us improve our work, below you will find answers to all your comments in blue.

Reviewer 1

The manuscript “Can exposure to environmental pollutants be associated with less effective chemotherapy in cancer?” provides an interesting overview of the literature establishing a possible link between environmental pollutants and chemotherapy resistance. The text is clearly written.

Q1.

It would be important that the authors clarify why would one look for the link between environmental pollutants and chemotherapy resistance. A clear hypothesis for the analysis presented in this manuscript would help the reader to understand the reasoning behind this extensive literature review.

R1. We add a paragraph at the end of the introduction to the manuscript to explain our hypothesis. In the manuscript they appear as follows:

Our hypothesis is that environmental contaminants could affect the effectiveness of chemotherapeutic treatments, even at concentrations below those established by current regulations, and thus negatively affect the clinical prognosis of cancer patients. In this way, the objective of this work is to show evidence that suggests that environmental pollutants can reduce the efficacy of chemotherapeutic drugs (Table 1), with a special emphasis on explaining the possible underlying molecular mechanisms that lead to the generation of this chemoresistance.

Q2.

In addition, the authors highlight some specific chemoresistance mechanisms that the authors “consider relevant” (line 129), but this should be better explained. The pathways discussed within each of these specific chemoresistance mechanisms are limited, taking into account the vast literature for each on of these. For example in the topic “Activation of oncogenic signaling pathways”, keys pathways/key factors such as KRAS or EGFR. In particular the first one has been linked to NRF2 activation, mentioned by the authors (PMID: 30824678). Similarly in the DNA repair topic, some key pathways such as NHEJ are missing (see for example PMID: 33628188) or for the xenobiotics metabolism (see for example PMID: 26180516). Also it would be important to discuss the role of extracellular matrix (ECM), both as part of tumor microenvironment and EMT. ECM plays a critical role in chemotherapy resistance (PMID: 32118030), being also affected by environmental pollutants (PMID: 31354480). Moreover it can have implications for other cell mechanisms that promote chemotherapy resistance, such as DNA repair, oxidative stress or cell signalling (PMID: 34394183, PMID: 34733852). Specific ECM alterations occur in response to environmental pollutants and examples should be included in the discussion part.

R2. As you recommended, we expand the information in the section "Molecular mechanisms of chemoresistance in cancer" adding the references you recommended and some more. We added a subsection called "Extracellular Matrix" and discussed ECM alterations that occur in response to environmental contaminants. In the manuscript it appears as follows:

  1. Molecular mechanisms of chemoresistance in cancer

Currently, several molecular chemoresistance mechanisms have been identified, which mainly include an enhanced efflux of drugs, oncogenic signaling pathways, apoptosis avoidance, tumor microenvironment signals, down-regulation of tumor suppressor genes, mitochondrial alteration, activation of antioxidant enzymes and scavengers, in-creased DNA repair capacity, increased metabolism of xenobiotics, autophagy, EMT, and an increase in cancer stem cells (CSCs). Notably, more than one mechanism can be used by the cancer cell to generate chemoresistance [34,35]. Below, we give a brief description of those that we consider relevant to better understand the following parts of this review.

4.1. Enhanced drug efflux

One of the main causes of drug resistance is increased drug efflux by ATP-binding cassette (ABC)-superfamily multidrug efflux pumps, which often consist of multiple subunits and whose function is to translocate substrates through the membrane against the concen-tration gradient using the energy released by the hydrolysis of ATP in its subunits with ATPase activity. A total of 15 members of this protein family have been implicated in potentially conferring resistance to chemotherapeutic agents, but three of them are the most studied, the P-glycoprotein (P-gp/MDR1/ABCB1), multidrug resistance protein1 (MRP1/ABCC1), and breast cancer resistant protein (BCRP/ABCP/MXR) [36,37]. P-gp overexpression has been observed in different types of hematological and solid tumors, where it favors the discharge of drugs such as colchicine, tacrolimus, dexamethasone, steroids, quinidine (QUIN), etoposide (VP-16), DOX, and VIN [34]. Meanwhile, MRP1 prevents the intracellular accumulation of glutathione (GSH)-conjugated xenobiotics and GSH-conjugated metabolites, which also confers drug resistance in tumor cells treated with daunorubicin, methotrexate (MTX), DOX, VP-16, and VCR [38]. Ultimately, BCRP is overexpressed in breast cancer and has a negative correlation with intracellular levels of mitoxantrone, daunorubicin, topotecan, and DOX [34].

4.2. Activation of oncogenic signaling pathways

The PI3K/AKT pathway is an important signaling pathway for chemoresistance in a variety of cancers including breast cancer, leukemia, lung cancer, ovarian cancer, hepatocellular carcinoma, and melanoma. This pathway generates a survival signal to resist cytotoxic anticancer drugs and improves the characteristics of cancer stem cells (CSCs) [39]. The MAPK/ERK pathway is widely used by cancer cells subjected to envi-ronmental stresses such as chemotherapy and the antitumor activity of the host immune system. Chemoresistance occurs because this pathway cooperates with some factors of the tumor microenvironment, activates pro-survival pathways that lead to cell proliferation and migration, as well as with the positive regulation of drug efflux transporters and the modulation of apoptosis, differentiation and senescence [40]. Meanwhile, activation of the NF-κB pathway has been reported with various chemotherapeutic agents such as daunomycin, bortezomib, PTX, VIN, VCR, DOX, 5-FU, CDDP, and TAM. This pathway frequently contributes to the induction of chemoresistance and radioresistance through the regulation of several genes involved in immunoregulation, growth regulation, in-flammation, carcinogenesis, and apoptosis [41]. Notch signaling is frequently dereg-ulated in many cancers, most commonly by over-activation, and confers a survival ad-vantage on tumors. Recent studies show that Notch signaling favors chemoresistance by promoting the characteristics of CSCs and inducing EMT [42]. The NRF2-ARE pathway regulates GSH metabolism and the expression of enzymatic antioxidant systems and their cofactors to restore redox homeostasis [43]. Several studies have shown that cancer cells with high levels of NRF2 are less sensitive to etoposide, CDDP and DOX, since there are more antioxidants that counteract the ROS produced by these drugs [44]. Furthermore, NRF2 has also been associated with sustained proliferative signaling and insensitivity to anti-growth signals, resistance to apoptosis, sustained angiogenesis, tissue invasion and metastasis, metabolic reprogramming, and immune system evasion [43]. RAS is the most frequently mutated oncogene in human cancers, and KRAS is the most frequently mu-tated RAS isoform. The mutated RAS protein is constitutively activated and enhances signaling pathways that promote cell growth, proliferation, and survival, as well as mobility and migration [45]. Oncogenic KRAS has been reported to be capable of tran-scriptionally increasing NRF2 levels and enhancing chemoresistance [46].

4.3. Increased DNA repair capacity

Another possibility that tumor cells become resistant to a variety of anticancer drugs is their ability to repair DNA damage. Cancer cells can overcome DNA damage caused by chemotherapeutic drugs by activating several different repair mechanisms and pathways [47,48]. For example, if the repair pathway that is responsible for triggering cell death after chemotherapy therapy is deficient, an alternative repair pathway compensates and prompts cancer cells to repair the damage, leading to resistance to chemotherapy [49]. A significant correlation has been shown between overexpression of the XPF and ERCC-1 proteins of the nucleotide excision repair (NER) pathway and the development of CDDP resistance in cancer cells [35]. Furthermore, the repair and tolerance of these lesions also involves homologous recombination (HR) pathway proteins such as RAD51, MRE11, ATM, ATR, and BRCA1 [35,50]. Defects in the ATM-Chk2-p53 pathway also contributed to the radiation resistance of glioblastoma cells [51]. Meanwhile, DNA-PK activity, in-volved in double-strand repair through the non-homologous end joining (NHEJ) path-way, plays a role in chemoresistance and its inhibition enhances the sensitivity of cancer cells to different chemotherapeutic drugs [52]. O6-methylguanine-DNA methyltrans-ferase (MGMT) overexpression has contributed to acquired resistance to temozolomide (TMZ) and other alkylating agents [53]. Meantime, the process of mutagenic translesion synthesis (TLS), responsible for repair of inter-strand DNA cross-links, has also been significantly asso-ciated with the development of resistance in cancer cells [35].

4.4. Elevated xenobiotic metabolism

Most chemotherapeutic drugs are subject to being metabolized by cytochrome P450. Cytochrome isoenzymes such as CYP1A6, CYP1A2, CYP1B1, CYP2C9, CYP2B6, CYP2C19, CYP3A4/5, and CYP2D6 are essential for phase I drug metabolism and de-toxification [54]. CYP1B1 overexpression in various types of cancer has previously been reported to modify the biotransformation of chemotherapeutics, such as mitoxantrone, flutamide (FLUT), docetaxel (TXT), and PTX [55]. In addition, increased expression of the CYP2A6 enzyme, which is involved in the metabolism of ifosfamide (IFO), cyclo-phos-phamide (CTX), aflatoxin, and 5-FU, has been found in some tissues of chimeric resistant breast tumors [56]. Furthermore, the highly upregulated expression of CP4Z1, CYP1B1, and CYP2A7 in cancer cells was associated with their increased resistance to a variety of chemotherapeutic agents [57].

4.5. Increased CSCs

CSCs are a cellular subpopulation of cancer cells characterized by their ability to self-renew, multiple differentiation, drug resistance, and tumor formation and growth [34]. The chemoresistance of CSCs may be related to the fact that these cells maintain a qui-escent state, activate drug afflux mechanisms, have enhanced DNA repair mecha-nisms, and can acquire an EMT phenotype, among many other things [58]. So far, it is unknown whether the number of CSCs in a tumor increases as a result of selection due to chemotherapeutic agents or because conditions are propitiated for these cells to proliferate, or both [59]. However, when the number of CSCs increases, it is associated with a worse clinical prognosis, a more aggressive phenotype and chemoresistance [60].

4.6. Extracellular matrix (ECM)

The tumor microenvironment comprises several components that are not malignant by themselves, but that play a fundamental role in creating adequate conditions for the growth and sustainability of tumors, such as tumor vasculature, connective tissue, infil-trating immune cells and the extracellular matrix (ECM), among others [61]. ECM and its sequestered growth factors are a fundamental component for all cells, although its role in CSCs has recently been highlighted. Pathologic remodeling of the ECM is an established hallmark of cancer, and the ECM is a key mediator of metastasis and drug resistance [62]. Several mechanisms of chemoresistance involving the ECM have been identified in all types of cancer and have been classified into a variety of categories including physical barriers to treatment (hypoxia, pH, and interstitial fluid pressure), associated drug re-sistance with cell adhesion (ECM organization, mechanosignalling, and pro-survival signaling pathways) and the effect of ECM on subpopulations of inherent stem cells or CSCs specifically [61,62]. Furthermore, ECM may also have implications for other cellular mechanisms that promote resistance to chemotherapy, such as DNA repair and oxidative stress, among many others [63,64].

Specific comments

Q3

- The headings Material and Methods, Results and Discussion should be replaced but other more specific headings such as “chemotherapeutic agents”, “chemoresistance mechanisms” and . The headings in their current form are not related with the content of each section

R3. You are right, this happened because we sent the manuscript in a free format and the editorial production did not properly copy the subtitles. We have placed the subtitles as they correspond.

Q4.

- Table 1 – Legend for the letters “a”, “b” and “c” is missing

  1. You are right, this happened because we sent the manuscript in a free format and the editorial production did not properly copy the subtitles. We add the meanings of “a”, “b” and “c” in the manuscript. In the manuscript they appear as follows:

a Per cubic meter, b Kg body weight/day, c European chemical agency (ECHA) https://echa.europa.eu,d European environment agency (EEA) https://www.eea.europa.eu

Q5.

- Lines 91-93 – the mode of action of DOX may be through direct TopoII targeting or ROS generation. In addition the role of TopoII os much broader then what has been described

R5. As you suggest, we add more information on the mechanisms of how DOX works to kill cancer cells. In the manuscript they appear as follows:

DOX is classified as an anthracycline antibiotic and is commonly used to treat some hematologic malignancies such as leukemias and Hodgkin's lymphoma, as well as solid tumors such as cancers of the bladder, breast, stomach, lung, ovaries, thyroid, soft tissue sarcoma, and others. Its main mech-anism of action is through its intercalation in DNA and the disruption of topoisomerase-II-mediated DNA repair [26]. In addition, DOX can also induce ROS production, damage mitochondrial DNA, disrupt major mitochondrial functions, and reduce membrane potential with consequent release of cytochrome C and induction of apoptosis, among other things [27]

Q6.

- Lines 122-123 – the sentence “identified, but mainly these include” could be changed into “identified, which mainly include” The authors provide an extensive list of possible mechanisms

R6. This was fixed as you suggested.

Reviewer 2 Report

The manuscript submitted by Lagunas-Rangel and his colleagues, provide a review in the current understanding on the potential role of environmental pollutants in chemoresistance in cancer patients, the manuscript is interesting, well written and easy to understand and digest. Although some comments are recommended to improve the manuscript:

  1. The titles of sections and subsections, this is not a research article, so including section entitled material and methods, results and discussion is not applicable here, so authors should change the section titles in a way the goes with the flow of the review paper (suggestions could be,: instead of materials and methods the section title could be “ General mechanisms of chemotherapies in cancer”, instead of results, they may include the section title as “mechanism of chemoresistance in cancer”, instead of discussion they may include as section title as “ role of environmental pollutants in chemotherapy resistance in cancer”.
  2. It is recommended to add a paragraph (in addition to a flow chart) at the end of the introduction section to highlight the methods the authors used in this review (included what databases they used to search for article (PubMed, google scholar…etc.)? What keywords they used in their search? how many papers were initially retrieved? What were their inclusion and exclusion criteria? How many papers finally were included in this review? ...etc.
  3. The authors should include an abbreviations list at the end of the manuscript after the conclusion

Author Response

Dear Reviewer,

We appreciate all your comments that help us improve our work, below you will find answers to all your comments in blue.

The manuscript submitted by Lagunas-Rangel and his colleagues, provide a review in the current understanding on the potential role of environmental pollutants in chemoresistance in cancer patients, the manuscript is interesting, well written and easy to understand and digest. Although some comments are recommended to improve the manuscript:

Q1.

  1. The titles of sections and subsections, this is not a research article, so including section entitled material and methods, results and discussion is not applicable here, so authors should change the section titles in a way the goes with the flow of the review paper (suggestions could be,: instead of materials and methods the section title could be “ General mechanisms of chemotherapies in cancer”, instead of results, they may include the section title as “mechanism of chemoresistance in cancer”, instead of discussion they may include as section title as “ role of environmental pollutants in chemotherapy resistance in cancer”.

R1. You are right, this happened because we sent the manuscript in a free format and the editorial production did not properly copy the subtitles. We have placed the subtitles as they correspond.

Q2.

  1. It is recommended to add a paragraph (in addition to a flow chart) at the end of the introduction section to highlight the methods the authors used in this review (included what databases they used to search for article (PubMed, google scholar…etc.)? What keywords they used in their search? how many papers were initially retrieved? What were their inclusion and exclusion criteria? How many papers finally were included in this review? ...etc.

R2. We added a section and flowchart based on the PRISMA 2020 rules. In the manuscript they appear as follows:

  1. Methods

This review is based on evidence collected by performing a PubMed and Google Scholar query using the words "cancer AND resistance AND pollutants" as search terms. The search strategy was implemented by manually searching the references reported by the most relevant studies on this topic. Figure 1 shows a flowchart of the steps that were followed to find evidence that contaminants could affect the effectiveness of chemotherapy drugs, as well the number of articles excluded and included in each step based on the Preferred Reporting Items for Systematic Reviews and Meta-Analyses (PRISMA 2020).

Figure 1. Flowchart describing the protocol adopted in this review based on PRISMA 2020

Q3.

  1. The authors should include an abbreviations list at the end of the manuscript after the conclusion

R3. We add the list of abbreviations as you suggested. In the manuscript they appear as follows:

Abbreviations: DN(M)EL: Derived no-effect or minimum effect level, BPA: Bisphenol A, POP: Persistent organic pollutant, BaP: Benzo[a]pyrene, AlCl3: Aluminum chloride, PCB-1254: Poly-chlorinated biphenyls, HBCD: Hexabromocyclododecane, OP: 4-tert-octylphenol, DEHP: Bis (2-ethylhexyl) phthalate, PAH: Polycyclic aromatic hydrocarbon, CDDP: Cisplatin, DOX: Doxo-rubicin, VIN: Vinblastine, 5-FU: 5-fluorouracil, PTX: Paclitaxel, CPT: Camptothecin, HCC: Hepa-tocellular carcinoma, BC: Breast cancer, COAD: Colon adenocarcinoma, OC: Oesophageal cancer, SCC: Squamous cell carcinoma, MEF: Mouse embryonic fibroblasts, CNM: Concentration not mentioned, EMT: Epithelial-to-mesenchymal transition, CSC: Cancer stem cell, ABC: ATP-binding cassette, QUIN: Quinidine, VP-16: Etoposide, MTX: Methotrexate, TMZ: Temozolomide, FLUT: Flutamide, TXT: Docetaxel, IFO: Ifosfamide, CTX: Cyclophosphamide, VIN: Vinblastine, GSH: Glutathione, NER: Nucleotide excision repair, HR: Homologous recombination, MGMT: O6-methylguanine-DNA methyltransferase, TLS: Translesion synthesis, OCP: Organochlorine pesticides, PBDE: Polybrominated diphenyl ethers, PFOS: Perfluorooctane sulfonate, PCDD: Poly-chlorinated dibenzo-p-dioxins, PCDF: Polychlorinated dibenzofurans, HBCD: Hexabromocy-clododecane.

Round 2

Reviewer 2 Report

I would like to thank the authors for addressing  all my suggestions

This manuscript is a resubmission of an earlier submission. The following is a list of the peer review reports and author responses from that submission.